# On the Dynamics of the Carbon–Bromine Bond Dissociation in the 1-Bromo-2-Methylnaphthalene Radical Anion

**DOI:** 10.3390/molecules27144539

**Published:** 2022-07-15

**Authors:** Marco Bonechi, Walter Giurlani, Massimo Innocenti, Dario Pasini, Suryakant Mishra, Roberto Giovanardi, Claudio Fontanesi

**Affiliations:** 1Department of Chemistry “Ugo Schiff”, University of Florence, Via della Lastruccia 3, 50019 Sesto Fiorentino, Italy; marco.bonechi@unifi.it (M.B.); walter.giurlani@unifi.it (W.G.); 2National Interuniversity Consortium of Materials Science and Technology (INSTM), Via G. Giusti 9, 50121 Firenze, Italy; 3Center for Colloid and Surface Science (CSGI), Via della Lastruccia 3, 50019 Sesto Fiorentino, Italy; 4Department of Chemistry, University of Pavia, Via Taramelli 10, 27100 Pavia, Italy; dario.pasini@unipv.it; 5Center for Integrated Nanotechnologies, Materials Physics and Applications Division, Los Alamos National Laboratory, Los Alamos, NM 87545, USA; mishra.498@gmail.com; 6Department of Engineering “Enzo Ferrari” (DIEF), University of Modena, Via Vivarelli 10, 41125 Modena, Italy; roberto.giovanardi@unimore.it

**Keywords:** radical anion, potential energy surface, dissociation, DFT, electron affinity, DRC, molecular dynamics, 1-bromo-2-methylnaphthalene

## Abstract

This paper studies the mechanism of electrochemically induced carbon–bromine dissociation in 1-Br-2-methylnaphalene in the reduction regime. In particular, the bond dissociation of the relevant radical anion is disassembled at a molecular level, exploiting quantum mechanical calculations including steady-state, equilibrium and dissociation dynamics via dynamic reaction coordinate (DRC) calculations. DRC is a molecular-dynamic-based calculation relying on an ab initio potential surface. This is to achieve a detailed picture of the dissociation process in an elementary molecular detail. From a thermodynamic point of view, all the reaction paths examined are energetically feasible. The obtained results suggest that the carbon halogen bond dissociates following the first electron uptake follow a stepwise mechanism. Indeed, the formation of the bromide anion and an organic radical occurs. The latter reacts to form a binaphthalene intrinsically chiral dimer. This paper is respectfully dedicated to Professors Anny Jutand and Christian Amatore for their outstanding contribution in the field of electrochemical catalysis and electrosynthesis.

## 1. Introduction

Electrochemistry is a powerful tool to study organic reactivity for the characterization of molecular properties as well as electrochemically driven synthesis. The latter case involves the formation or dissociation of chemical bonds, in general through the production of radical species yielded upon electrochemical oxidation or reduction. Within this field, the electroreductive cleavage of the carbon–halogen bond in halogenated organic compounds is a widely studied topic, since it is a fundamental reaction common to a wide variety of synthetic applications in organic electrochemistry [1,2]. Moreover, the possibility of removing the halogen group from organic substrates has been considered for environmental applications (electrochemical incineration) [3,4]. Several organic halocompounds are known to have an environmental impact due to their use as herbicides or cleaning agents [5,6]. Although many of these compounds are no longer commercially available, recent studies proved the presence of volatile organic halides in connection with residential and non-residential activities [7]. For these reasons, fundamental research concerning the selective removal of the halogen group represents an important area interest for environmental applications. Nowadays, the use of electrochemistry is considered a useful tool to produce specific products at an industrial scale [8,9]. Indeed, electrochemical reactions can be implemented under favorable conditions, especially from an energy and environmental point of view [10,11,12]. The electroreduction in organic halides has been a key topic in organic electrochemistry studies, which coupled the voltametric investigation to the synthetic potentialities of controlled potential electrolysis [13]. In the past, radicals were considered too reactive to be used in organic synthesis, but today, it is clear that radicals often offer a higher level of selectivity and predictability than conventional ionic reactions [14]. In fact, it is, in principle, feasible to create radicals using electrochemical methods to promote a high degree of regio-, chemo- and stereoselectivity. Interesting results could also be achieved by applying the spin-dependent electrochemistry paradigm to the reduction in organic halocompounds. In fact, it is theoretically possible to apply the spin-filtering effect exhibited by particular electrodes to achieve asymmetric syntheses [15]. Thus, this paper focuses on the 1-bromo-2-methylnaphthalene reduction mechanism, which we can consider a “case study” reaction for the electrochemical reduction in halogenated organic compounds. The latter is quite an important chemical reaction, also in connection with applicative aspects connected to the degradation of environmental pollutants and the synthesis of graphene-like films (by repeated covalent coupling of aryl radicals) [16]. In fact, 1-bromo-2-methylnaphthalene reduction must involve the formation of radical species, and the electron transfer could induce the dissociation of the closed-shell halide anion [17,18]. The electrochemical reduction in the neutral 1-bromo-2-methylnaphthalene species leads to the formation of a radical anion, which is obtained by the injection of one electron [19]. A general model for the electrochemical reduction in halogenated naphthalene derivatives has been proposed by Jean-Michel Savéant, which distinguishes two types of a reaction path: (i) stepwise or (ii) concerted mechanism [20,21,22]. It is typically considered a “stepwise” mechanism where the intermediate radical anion species is formed, even in the case of an extremely short-lived species [23]. The first elementary reaction in the stepwise mechanism is the electron transfer from the electrode to the organic halide that causes the formation of an intermediate radical anion. The radical anion formed in the first step subsequently undergoes a cleavage of the C−X bond with the formation of the bromide anion and an organic radical. On the other hand, the concerted mechanism involves direct production of a bromide anion and an organic radical without the formation of any intermediate state. The existence of a stepwise or concerted reaction path depends on different parameters, such as the nature of the investigated halide, the solvent and the influence of the supporting electrolyte [24]. In the scenario of aromatic halides, the stepwise reaction is, in general, the preferred dissociation pathway [25]. Based on these considerations, we propose the reaction path shown in Figure 1. In our study, we focused on the initial reaction steps leading to the formation of a closed-shell radical species. Dimers are then produced by a reaction between radicals, following the formation of a C−C bond. The polymerization reaction proceeds via the dissociation of a C−Br bond of the 1-bromo-2-methylnaphthalene. The organic radical produced during the reaction pathway could react with another radical to form a dimer. At extremely negative potentials, the organic radical can be further reduced to form the anion of the de-halogenated organic compound. This anion can bind with a proton from the solvent or residual water. In this paper, a theoretical approach is followed to shed light on the reduction elementary mechanism, in detail which is experimentally out of reach. The electronic structure of the radical anion produced by the electron uptake is analyzed by a suitable selection of a π* or σ* molecular orbital configuration. The reduction mechanism relies on an ab initio steady-state potential energy surface analysis and on molecular dynamics (MD) calculations by using the dynamic reaction coordinate (DRC) method. It is clearly shown that electron transfer induces cleavage of the carbon–bromine bond and that the organic radicals formed can react to form a stable closed-shell dimer.

## 2. Results and Discussion

### 2.1. The Mechanism

The common-base molecular framework consists of 2-methylnaphthalene with a halogen substituent being (Br) in 1-position: 1-bromo-2-methylnaphthalene (**1**). Figure 1 shows the single elementary steps considered in the electroreduction in 1-bromo-2-methylnaphthalene (**1**). The first two elementary steps are mainly inspired by the Savéant mechanism, while the last step represents the possible formation of a chiral closed-shell dimer [26,27]. The elementary reaction steps of three different reaction paths are considered. Path A refers to the simple one-electron reduction in 1-bromo-2-methylnaphthalene (**1**), leading to the relevant radical anion (**2**); in principle, this is a reversible step. Suitable experimental conditions, low temperature, very fast potential scan in cyclic voltammetry (CV) and ultradry and oxygen-free solvent could allow for the observation of a reversible or quasi-reversible voltammogram [28]. Appendix A shows the potential energy surface for reaction path A. Path B involve the dissociation of the C−Br bond, leading to the neutral radical species (**3**), which is a crucial reactive species in the possible propagation mechanism for the formation of chiral dimers. The simplest follow-up reaction involves two radicals (**3**); path C yields a stable closed-shell dimer. The molecular framework of the latter consists of 1,1′-binaphthyl with a methyl substituent in the 2-position: 1,1′-binaphthalene, 2,2′-dimethyl. 1,1′-binaphthalene, 2,2′-dimethyl exists in two different enantiomeric forms, R and S, referred to in this work as (**4**) and (**4′**). Even if these molecules do not show chirality centers, they are still chiral. This is because the rotation around the C−C bond linking the two naphthalene units is obstructed by the presence of the methyl substituents.

### 2.2. Reaction Path B

As shown in Figure 1 (relaxed-scan PES of the elementary reaction $\left(2\right)\to \left(3\right)+{\mathrm{Br}}^{-}$), reaction path B is a downhill energy process. The transition state of path B TS(B) is shown in the right panel in Figure 1. The latter is found to be 2 kcal mol^−1^ lower in energy than the reagents; thus, the dissociation process of the bromide anion has no activation energy. As is also shown in Figure 2 (σ radical, black squares), after the first electron uptake, the carbon halogen bond dissociates following a downhill energy path with no activation energy. This means that most anion radicals (**2**) dissociate to form the species (**3**). Reactions of this kind are known in the literature as anti-Arrhenius processes [29]. Considering the thermodynamic point of view, the products of path B $\left(3\right)+{\mathrm{Br}}^{-}$ are 32 kcal mol^−1^ more stable in energy than the reagents (**2**).

Figure 2 sets out the potential energy curves of 1-bromo-2-methylnaphthalene radical anion (**2**) as a function of the carbon–bromine bond distance. The dissociation process of the C−Br bond depends strictly on the type of electronic configuration considered for the radical anion: doublet ^2^$A^{\prime }$ (σ radical) or ^2^$A^{\prime \prime }$ (π radical) electronic states. The ^2^$A^{\prime \prime }$ (π radical, red circles) potential energy curve vs. C−Br bond distance features a “Morse-like” pattern of associative nature. A minimum is present at $d_{C-br}=1.96\;Å$ (optimized geometry at the UB3LYP/cc-pVTZ level of the theory, Cs symmetry). The dissociation is predicted to be a global up-hill in the energy process for the ^2^$A^{\prime \prime }$ electronic state. On the contrary, the ^2^$A^{\prime }$ (σ radical) electronic state (black squares, Figure 2a) shows a decreasing pattern as a function of the C−Br bond distance. At the C−Br bond distance $d_{C-Br}=2.12\;Å$, the two electronic states are degenerate in energy. Figure 2c,d show the molecular orbital density distribution of the π and σ type radical anions. For the sake of comparison, Appendix A shows the unrelaxed scan PES, PM3 level of the theory.

Appendix A show the potential energy curves as a function of the carbon–bromine bond distance at the UB3LYP/cc-pVTZ and PM3 level of theory for the neutral molecule (**1**). In this case, the curve is associative, and the PES pattern closely resembles a Morse-like behavior. This evidence confirms that electronic transfer is a key step in the dissociation of the C−Br bond, as the neutral 1-bromo-2-methylnaphthalene shows an upward trend in the energy-dissociative pathway (the dissociation energy is larger than 50 kJ mol^−1^, compare Appendix A).

### 2.3. Path C

Path C represents the progression of products obtained via path B. Path C focuses on the reaction between two 2-methylnaphthalene radical anions (**3**). The energy pattern of path C is shown in Figure 3, where we can observe a PES with three stationary-state systems: (i) reagents $\left(3\right)+\left(3\right)$; (ii) transition state $TS\left(C\right)$; and (iii) product (**4**) or (**4′**). Figure 3 also shows the transition states’ TS(C) molecular structure. Route C features the direct coupling of radical cations, (**3**) to produce (R)-1,1′-binaphthalene, 2,2′-dimethyl or (S)-1,1′-binaphthalene, 2,2′-dimethyl, in this study denoted by (**4**) and (**4′**). Hessian analysis of products (**4**) and (**4′**) shows that they are a real minimum of the PES; indeed, all frequencies of these structures are real and positive. Like path B, path C is a monotonic downhill energy process, and no activation energy is present. 

The neutral closed-shell dimer in Figure 4 shows a minimum potential energy for a bonding distance between the C−C bond, linking the two naphthalene units at about 1.5 Å. Appendix A shows the molecular potential energy vs. $C_{1}-C_{2}$ distance for the neutral closed-shell dimer 1,1′-binaphthalene, 2,2′-dimethyl (**4**), unrelaxed potential energy surface for the C–C association (UB3LYP/cc-pVTZ level of theory) vs. relaxed potential energy surface. 

A markedly different picture is obtained if we consider the molecular potential energy vs. $C_{1}-C_{2}$ distance for the dimer 1,1′-binaphthalene, 2,2′-dimethyl (**4**) in the triplet state (compare Appendix A). In this case, a higher potential energy curve is observed than that observed for the single state (Figure 4). As mentioned above, even if the dimer does not show chirality centers, it is still chiral. This is because the rotation around the C−C bond linking the two naphthalene units is obstructed by the presence of the methyl substituents. This hypothesis was verified by calculating the molecular potential energy vs. the $C_{1}-C_{2}$ dihedral angle of the dimer. The potential energy curve shown in Appendix A presents a minimum energy for a dihedral angle of 90 degrees. A complete rotation corresponding to the conversion of the enantiomer R into the enantiomer S is not possible, as it requires more than 80 kcal mol^−1^. 

### 2.4. Molecular Dynamics: Path B and C DRC

The dynamics of the electroreduction process is characterized by using the dynamic reaction coordinate (DRC) algorithm [30], which is a classical dynamic calculation based on an ab initio potential energy surface. The calculations of ab initio molecular dynamics (DRC) were performed starting from different equilibrium structures of the PES, such as reagents or transition states. The initial velocity vector and the initial DRC geometry are obtained by projection of the Hessian matrix obtained from a previous Hessian calculation; the kinetic energy is partitioned over each normal mode of the Hessian matrix.

Figure 5 shows the results of a typical DRC calculation, B3LYP/6-31G(d) level of theory, of the 1-bromo-2-methylnaphthalene radical anion (**2**). The energy as a function of time, Figure 5a, shows a marked ripple, which is due to the activity of the intramolecular/intermolecular vibrational mode, while Figure 5b displays the variation of the C−Br bond distance as a function of time. Remarkably, there is a sudden, almost vertical increase in the 190 to 500 fs time interval. This corresponds to a sort of discontinuity in the *energy* vs. *time* curve, which is followed by a decrease in the average energy of about 40 kJ mol^−1^. A completely different figure is obtained by considering neutral 1-bromo-2-methyl naphthalene (before electron transfer). In this case, shown in Appendix A, DRC trajectories are obtained in which the bond distances oscillate between 1.85 and 2.10 Å, so a neat dissociation of the C−Br bond does not occur. Once again, it is shown that electronic transfer is a key step in the dissociation of the carbon–halogen bond. The same conclusions are drawn using the PM3 level of theory (compare Appendix A). 

Figure 6 shows the DRC results of two 1-methylnaphthalene (**3**) radicals forming 1,1′-binaphthalene, 2,2′-dimethyl (**4**), considering a single total multiplicity for the system, i.e., two radicals of type alpha and beta, respectively. ${\left(3\right)}^{\alpha \cdot }+{\left(3\right)}^{\beta \cdot }\to \left(4\right)$. Figure 6a (energy as a function of time) shows a marked ripple, which is due to the intramolecular/intermolecular vibrational mode, while Figure 6b displays the variation of the C−C bond distance between the two aromatic rings as a function of time. Remarkably, there is a sudden, almost vertical increase in the 0 to 200 fs time interval. This corresponds to a sort of discontinuity in the *energy* vs. *time* curve, which is followed by a decrease in the average energy of about 90 kcal mol^−1^. The same results were obtained by starting the DRC calculation from a different distance between the two radicals (Appendix A). The DRC results confirm that following the dissociation of the bromide anion from the anion radical (**2**) forming the organic radical (step B), the reaction between the two radicals (step C) is obtained.

A completely different picture is obtained by considering a system featuring a triple total multiplicity, i.e., two radicals of the same type, both alpha and beta. ${\left(3\right)}^{\alpha \cdot }+{\left(3\right)}^{\alpha \cdot }\to \left(4\right)$. In this case, shown in Appendix A, DRC trajectories are observed in which the bonding distance becomes longer as time increases from 4.0 to 6.0 Å, thus hindering bond formation between the two radicals. 

To suggest possible methods useful for monitoring the time evolution of the electrochemical reaction experimentally, i.e., by conducting a spectro-electrochemical in situ experiment, the theoretical UV-VIS, IR and CD spectra of all the species under study (**1**)–(**4**) were calculated. Appendix A shows the UV-Vis spectra calculated at the UB3LYP/cc-pVTZ level of theory, in the wavelength range between 200 and 650 nm. Various absorption peaks are observed depending on the species considered, especially for the radical anion (**2**), which shows an absorption peak at 610 nm, a wavelength at which the other species involved in the process do not absorb. Therefore, in principle, the collection of UV-Vis spectra would allow the in situ characterization of the species produced during electrochemical reduction. Appendix A shows the IR spectra calculated at the same level of theory, in which case, the spectra of the various species involved in the reaction mechanism are quite similar. Of course, asymmetry induction could be measured by measuring DC spectra as the two different enantiomers produced by the dimerization of the naphthalene radicals (4)-R and (4)-S, which could instead be identified by the opposite circular dichroism behavior. Appendix A clearly shows the difference in the absorption bands of the two molecules due to the Cotton effect.

## 3. Methods

Results were derived with ab initio quantum-mechanical-based methods, and all calculations were performed using C1 symmetry and unrestricted wave function [31,32,33]. Calculations were performed using Gaussian [34] and Firefly Rev 8.20 [35] programs. To view the molecular structures and molecular orbitals, Chemcraft [36] was used, while to display DRC trajectories, we used MacMolPlt [37]. To extract the molecular geometrical parameters from DRC calculations, we used original Fortran-based codes. The molecular geometries of the PES vs. reaction coordinates were obtained by full optimization carried out at UB3LYP/cc-pVTZ [38,39,40,41] levels of theory. Geometry optimization was carried out by using Barone and Cossi’s polarizable conductor model (CPCM) [42] to account for the solute−solvent (acetonitrile) interaction [32]. Hessian calculations (vibrational frequency spectrum) were used to check the stability of all investigated species. All the frequency values of reagents and products were found as real and positive, while for transition states, we found a single imaginary negative frequency. Dynamic reaction coordinate (DRC) trajectories (classical dynamic calculation based on an ab initio molecular PES) at the B3LYP/6-31G(d) level of theory were performed using GAMESS [43] and Firefly programs [35], starting from the molecular geometry of stationary points of the PES [44,45,46,47,48,49]. In DRC calculations, the velocity vector needed to start the calculations is obtained from Hessian vibrational eigenvectors, and the kinetic energy is partitioned over all normal modes of the species investigated, assigning the zero-point energy to each normal mode. The dissociation dynamics, via dynamic reaction coordinate (DRC) calculations, were performed in vacuum. In the case of PES vs. reaction coordinates and DRC trajectories, UB3LYP/6-31G(d) levels of theory (Appendix A) and semi-empirical PM3 calculations [50,51] were used for the sake of theoretical cross check. UV-VIS, IR and CD spectra of all the species under study (**1**)–(**4′**) were performed at UB3LYP/cc-pVTZ levels of theory.

## 4. Conclusions

In this paper, we theoretically modeled the dissociative electroreduction in 1-bromo-2-methylnaphthalene. The dissociation mechanism was disassembled at a molecular level and studied, exploiting the DFT ab initio analysis by calculating the reaction elementary steps: PESs. DRC calculations (classical molecular dynamics calculation based on an ab initio PES) were used to shed light on both the intramolecular and intermolecular geometrical variations as a function of time, aiming to describe the progress of the C−Br bond dissociation process at a molecular level. On the whole, a stepwise mechanism for the electroreduction dissociation of the C−Br bond was proposed. 

The main conclusions follow.

(1)The one-electron reduction in 1-bromo-2-methylnaphthalene yields a transient radical anion. The latter subsequently undergoes a cleavage of the C−Br bond with the formation of the bromide anion and an organic radical.

(2)Indeed, the neutral radicals produced following the dissociation of the radical anion parent species react to form a 1,1′-binaphthalene, 2,2′-dimethyl compound, an intrinsically chiral dimer.(3)UV-VIS, IR and CD spectra calculated at the UB3LYP/cc-pVTZ level of theory suggest possible experimental methods for monitoring the time evolution of the electrochemical reaction, i.e., by conducting a spectro-electrochemical in situ experiment.

The work provides interesting information about redox reaction investigations of aromatic compounds. The theoretical investigation showed interesting properties of 1-bromo-2-methylnaphthalene that make it a suitable target molecule for future electrochemical studies. This approach is meant to provide precious information needed to design and select molecular architectures suitably tailored for cross-coupling polymerization.

## Data Availability

The data presented in this study are available in the Appendix A.

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
