# Peer review of "On the Dynamics of the Carbon–Bromine Bond Dissociation in the 1-Bromo-2-Methylnaphthalene Radical Anion"

_molecules, 2022, doi:10.3390/molecules27144539_

Round 1

Reviewer 1 Report

Work entitled "On the Dynamics of the carbon-bromine bond dissociation in 2 the 1-Bromo-2-Methylnaphthalene radical anion" is interesting. It discusses the electrochemical reduction of halogenated organic compounds by studying carbon-bromine bond reduction in 1-bromo-2-methylnaphthalene. However, the authors need to make some significant improvements in the manuscript.

Comments

1. The author should discuss the potential outcomes of the study.

2. Abstract should include the quantitative results obtained from the work.

3. Mention the full form of abbreviation use in Scheme 1.

4. In results and discussion, it is mentioned that path A of Scheme 1 is reversible but in Scheme 1 it is not showed. Make a correction.

5. In manuscript, figure S1 is mentioned in results and discussion, but it is not present in the whole manuscript. Clearly mention the figures S3, S4, S5, S6, S7, S8, S9, S10, S11, S12, S13 and S14 as well.

6. Give proper headings of different paths (A, B,) of the mechanism. Headings of path B and C are given while that of path A is missing.

7. In the manuscript under the heading of mechanism it is discussed that path A is reversible so use sign of reversible arrow in the chemistry of path A.

8. What is PES scan?

9. Clearly mention the figures S1 to S14 which are discussed in the manuscript.

10. TC (C) is discussed in the path C and caption of figure 3 but it is not shown in the figure 3.

Reviewer 2 Report

In this work, Bonechi and coworkers presented a theoretical investigation on the carbon-bromine bond dissociation in 1-Bromo-2-Methylnaphthalene radical anion by extensive calculations at dft and semi empirical level. The authors also performed dynamics simulations on the dissociation process. By carefully reading, I found the work very interesting and may be publishable, but the following should be addressed before the manuscript can be considered further.

1.         In the abstract, DRC was not defined before usage. It’s also confusing to use molecular dynamics and DRC together. The differences between DRC and molecular dynamics can be added to method section.

2.         What’s the solvent used in this work? How many solvent molecules were considered in the DRC simulations?

3.         The PES for path B looks odd. Why is the energy of transition state lower in energy than the reagents? Please discuss this in the context with additional proof or citations to relevant examples.

4.         Figure 2, right panel should be resized to match the one in the left panel.

5.         Also in Figure 2, it’s not clear which kind of radical would be plausible, how it will evolve with the C-Br distance, and what the barrier for this dissociation is. Please address in the context.

6.         The same problem was also found for path C for the combination of 2 radicals into a dimer that the transition state is lower in energy than the reactants.

7.         Figure 4, left panel, it would be more interesting to see the data for the relaxed potential energy surface for the C-C association.

8.         For Figure 5, would it be possible to identify the radical type for the one formed on the trajectory shown? It’s sad to see that the DRC simulations were performed with 6-31g(d). Would the result be scientifically consistent with those obtained with cc-PVTZ ? How can the energy change be correlated with the C-Br distance or other structural changes? Please justify in the context. Please indicate the dihedral angle in the structures shown in the upper panel.

9.         On lines 215-216, the authors mentioned that “DRC results confirm that following the dissociation of the bromide anion from the anion radical (2) forming the organic radical (step B), the reaction between two radicals (step C) is obtained.” Please explain how this was realized in the DRC calculations and add relevant evidence to the context.

10.     The scale for Figure 6c should be enlarged to show the complete distribution of angles.

Round 2

Reviewer 2 Report

As the authors have fully addressed my concerns in this revision, I suggest acceptance of the manuscript for publication in molecules.

Reviewer 3 Report

I have no more comments.